# Three-Year Patency Results following Endovascular Transvenous Femoropopliteal Bypass

**DOI:** 10.3390/medicina59030462

**Published:** 2023-02-25

**Authors:** Roberts Rumba, Dainis Krievins, Janis Savlovskis, Natalija Ezite, Aigars Lacis, Eva Petrosina, Ludovic Mouttet, Janis Gardovskis, Christopher K. Zarins

**Affiliations:** 1Vascular Surgery Department, Pauls Stradins Clinical University Hospital, Riga Stradins University, LV-1002 Riga, Latvia; 2Vascular Surgery Department, Pauls Stradins Clinical University Hospital, University of Latvia, LV-1002 Riga, Latvia; 3Diagnostic and Interventional Radiology Centre, Pauls Stradins Clinical University Hospital, LV-1002 Riga, Latvia; 4Statistics Laboratory, Riga Stradins University, LV-1002 Riga, Latvia; 5Faculty of Medicine and Health Sciences, McGill University, Montreal, QC H3A 0G4, Canada; 6General Surgery Department, Pauls Stradins Clinical University Hospital, Riga Stradins University, LV-1002 Riga, Latvia; 7Department of Surgery, Stanford University Medical Center, Stanford, CA 94305, USA

**Keywords:** transvenous femoropopliteal bypass, patency data, endovascular femoropopliteal bypass

## Abstract

*Background and Objectives*: Peripheral artery disease is one of the most common vascular pathologies. There is an ongoing debate among specialists on whether open or endovascular revascularization is preferred in cases of complex superficial femoral artery (SFA) lesions. The purpose of this study was to assess patency results of a relatively new transvenous endovascular bypass device. This could add to existing evidence and aid in comparison between open and endovascular bypass. *Materials and Methods*: Patients with complex TASC-C and D SFA lesions who had indications for revascularization were identified. Prospective analysis of stent graft patency from 54 transvenous femoropopliteal bypass procedures was performed. Patency was assessed by Duplex ultrasound every six months. Kaplan–Meier analysis was performed to assess primary, primary-assisted, and secondary patency of transvenous bypass. *Results*: Following endovascular transvenous femoropopliteal bypass, 3-year graft primary, primary-assisted, and secondary patency was 43.8%, 66.3%, and 73.9%, respectively. *Conclusions*: Transvenous endovascular femoropopliteal bypass is a viable option for selected patients who lack adequate saphenous vein or have comorbidities that increase the risk of open femoropopliteal bypass. Strict post-operative follow-up is necessary to improve patency rates.

## 1. Introduction

Atherosclerotic peripheral artery disease (PAD) is one of the most common conditions treated by a vascular surgeon and causes significant morbidity and mortality throughout the world. Within an aging population, PAD is only expected to increase in its prevalence, approximating 15% of the population above the age of 75 [1]. Despite this impressive prevalence, there is still an ongoing debate about the most effective revascularization technique for this elderly and comorbid patient population. While open femoropopliteal bypass was considered the gold-standard treatment for the previous four decades, endovascular revascularization has advanced rapidly and is currently competing with open surgery in various aspects [2]. To date, several randomized clinical trials have evaluated these two treatment options head-to-head, and the results have not strongly favored one method over the other [3]. The advantage of open surgery appears to be better long-term patency, lower secondary intervention rates, and 30-day major amputation rates for patients with tissue loss, whereas endovascular surgery provides lower 30-day mortality, morbidity, and hospital stay [4,5]. In an effort to combine the benefits of both methods, a new percutaneous endovascular transvenous femoropopliteal bypass technique (also known as the Detour procedure, invented by the company, PQ Bypass) was introduced. This relatively new approach appears to be safe with regards to the venous system [6,7], but the mid-term and long-term patency rates remain to be evaluated. Therefore, the focus of this study was to assess primary, primary-assisted, and secondary bypass patency after transvenous bypass to determine durability of the arterial conduit and the necessity for secondary procedures. Here we report our 36-month results.

## 2. Materials and Methods

In total, we included 52 subjects from 2015 until 2019 in the prospective, multi-center DETOUR trial at the Pauls Stradins Clinical University Hospital, Riga, Latvia. Two patients had bilateral superficial femoral artery (SFA) occlusions; therefore, 54 procedures were performed. One patient was lost to follow-up at the start of the study, one patient died 8 months after the procedure of an unrelated cause, and one patient did not attend follow-up after 24 months. Therefore, 51 procedures have complete 3-year data and 52 procedures have 24 months’ follow-up data for calculating Kaplan–Meier curves. In addition, several patients have longer follow-up data due to visits to the study center after the pre-defined 36-month period. The main inclusion criteria were SFA TASC C or D lesions with a patent popliteal and at least one crural artery, Rutherford class 3–5, and an ABI of 0.7 or less. The main exclusion criteria were a history of deep vein thrombosis, chronic kidney disease stages 4–5, Rutherford categories 0, 1, 2, and 6, and a history of major distal amputation in index or non-index leg. The deep venous system was investigated using Duplex ultrasound and only patients with popliteal vein diameter > 10mm (single or duplicate) were included in this study. Part of the standard of care in our center is to undergo three months of exercise therapy before considering surgical treatment for patients with claudication; therefore, only claudicants who failed conservative management were offered a surgical treatment. In addition, both open and endovascular treatment was discussed with patients. To avoid selection bias, endovascular transvenous bypass was only offered to patients who were otherwise considered and were amenable to open above-the-knee femoropopliteal bypass or conventional angioplasty. Patient selection for the Detour procedure is depicted in Figure 1. Approval by the ethics committee was obtained as well as a written consent to participate by all the patients included in the study.

To ensure consistency, all of these interventions were performed by a selected team of vascular surgeons and interventional radiologists. At the start of the procedure, patients received 5000 units of heparin with an additional dose as required to maintain an activated clotting time of 250–300 s. The contralateral common femoral artery was used for arterial access with 8Fr sheath and 0.035” guidewire, followed by crossover of the aortic bifurcation using a 0.014” guidewire, and placement of the PQ Crossing device (PQ Bypass, Milpitas, CA, USA) proximal to SFA occlusion. Using Duplex ultrasound (Canon Aplio i800, Tokyo, Japan) guidance and another 0.014” guidewire deep venous system, access was gained using ipsilateral crural or muscular vein through which a PQ Snare (PQ Bypass, Milpitas, CA, USA) device was introduced and positioned in the femoral vein opposite the PQ Crossing device. The Crossing device is equipped with a marker that is targeted towards the femoral vein using x-ray guidance. Knowing the position of this radio-opaque marker and the rotation of the angiography C-arm helps to direct the needle towards the targeted structure. A spring-loaded guidewire support and delivery system was then used to deploy a crossing needle through the walls of the SFA and femoral vein into the PQ Snare basket (PQ Bypass, Milpitas, CA, USA), which is used to snare a 0.014” guidewire. The next step is to withdraw both snare and guidewire from the body to create a through-and-through access from the contralateral femoral artery to the ipsilateral crural vein. Balloon dilatation was performed to create proximal anastomosis. Following this, the PQ Snare device was positioned in the femoral vein using 0.014” guidewire, the baskets were opened distally to the level or arterial occlusion, and the PQ Crossing device was advanced through proximal anastomosis in the femoral vein down to the level of the patent artery right next to it. The needle is once again deployed through the walls of the vein into the lumen of the patent artery using markers and x-ray guidance. A second 0.014” guidewire was then advanced through the needle into a patent popliteal artery. Balloon dilatation was performed to create distal anastomosis, typically using a 4 × 20 mm balloon. A stiffer 0.035” guidewire then replaces the 0.014”. As a final step, TORUS stent grafts (PQ Bypass, Milpitas, CA, USA; self-expanding nitinol wire encapsulated in ePTFE) were deployed in a distal to proximal fashion, creating a completely endovascular femoropopliteal above-the-knee bypass. A 6 cm overlap with adjacent stent graft is recommended. The majority of patients (59.3%) required two stent grafts. One patient received four stent grafts whereas the remaining 38.8% of patients required three.

After the procedure, the patients received a thorough follow-up with Duplex ultrasound at 6-month intervals to assess transvenous bypass patency, inflow, and outflow vessels, as well as deep venous vasculature. The patients were followed for 36 months after the procedure. Dual antiplatelet therapy (Aspirin 100 mg (G.L. Pharma GmbH, Lannach, Austria) + Clopidogrel 75 mg (Sanofi Winthrop Industrie, Ambarès & Lagrave, France)) was initiated starting from the procedure and continued throughout the 36 months’ follow-up. Exceptions were made in case the patient required anticoagulants due to cardiac arrythmia. Antiplatelet therapy was discontinued if clinically significant bleeding occurred, and was reinitiated as soon as safely possible. In addition, venous system anatomical and functional data were also documented using plethysmography. Venous system parameters and outcomes have been reported recently [6].

Primary patency was defined as time (in months) from initial restoration of vessel patency (index procedure) to any secondary intervention to sustain bypass patency. Primary-assisted patency was defined as time (in months) from initial procedure to impending failure of the bypass that was retreated during this time period due to a significant stenosis but not full thrombosis or occlusion of the graft. Secondary patency was defined as time (in months) from the primary procedure to complete bypass failure that was retreated during this time period due to complete occlusion of stent graft.

## 3. Results

### 3.1. Study Population

The mean age of the study population was 64.9 years (SD = 8.3). Most of the patients were male (92.3%) and had a mean baseline ABI of 0.62 (SD = 0.16). The prevalence of obesity was low with a mean body mass index of 27.5 (SD = 5.1). The majority of the patients had complaints of severe claudication (Rutherford 3 = 84.6%) and rest pain (Rutherford 4 = 9.6%). The most commonly observed comorbidities were arterial hypertension (88.4%, n = 46), smoking (83.3%, n = 45), and coronary artery disease (28.8%, n = 15). The mean follow-up period was 36 months, with the shortest and longest follow-up being 7 and 62 months, respectively. The study population and lesion parameters are summarized in Table 1.

### 3.2. Clinical Arterial Results

We assessed clinical improvements in the Rutherford class after surgery and compared these to the baseline. A significant reduction in symptoms and an increase in the Rutherford class by at least one category was observed in 96% of the patients one month after surgery (74% of the patients by 1 category and 22% of the patients by 2 or more categories). One year after surgery, 92.2% of the patients still maintained clinical improvements (76.5% of the patients by 1 and 15.7% by 2 or more). After 2 years of follow-up, 85.4% of the patients remained at least 1 category above the baseline Rutherford score, and at the 3-year follow-up, 73.2% still had improvements in their symptoms.

### 3.3. Primary Patency

Primary patency was 72.2% at 12 months, 55.6% at 24 months, and 43.8% at 36 months (Figure 2).

### 3.4. Primary-Assisted Patency

Patency was assisted by balloon angioplasty of a significant (>70%) stenosis in graft or SFA above the graft in most cases. Two cases had full detachment of stent grafts with arterial flow into the femoral vein and were retreated with a bridging stent graft 1 day and 1 month after the primary procedure (Figure 3). Primary-assisted patency was 88.9%, 77.8%, and 66.3% at 12 months, 24 months, and 36 months, respectively (see Figure 4).

### 3.5. Secondary Patency

Patency was restored by open, endovascular, or hybrid thrombectomy of a completely thrombosed stent graft. Twelve patients in total (23.5%, 12/51) developed full stent graft thrombosis during the 36-month follow-up period. Five of these cases had indications for repeated intervention due to severe claudication or rest pain. Following thrombectomy, patency in these cases was maintained for 16–34 months. Overall secondary patency was 92.6% at 12 months, 85.2% at 24 months, and 73.9% at 36 months (Figure 5). The patency data are summarized in Table 2.

## 4. Discussion

Saphenous vein is, undoubtedly, the best bypass material in the treatment of long and complex SFA lesions in terms of bypass durability, both above and below the knee. Reported 5-year patency for saphenous bypass is 82% compared to 26% for polytetrafluoroethylene (PTFE) [2]. In addition, saphenous vein bypass has the highest patency at 2 years compared to all other interventions, open or endovascular [8]. Using saphenous vein as a bypass is also associated with superior results in adverse limb events and death compared to endovascular revascularization in patients with chronic limb-threatening ischemia and adequate great saphenous vein, as recently reported in BEST-CLI trial results [9]. However, in 20–45% of patients requiring bypass surgery, greater saphenous vein is either inadequate or not available [10]. In these cases, either endovascular revascularization or open prosthetic graft bypass is used to treat SFA disease. Prosthetic grafts have a reported infection rate of up to 6% when anastomosis is located in the groin region [11]. Given the devastating consequences of vascular graft infections—30-day major amputation rate of 13% [12] and one-year mortality rate of 16% [13]—patients with tissue loss and other risk factors for graft infection might benefit more from endovascular revascularization in case of inadequate or unavailable saphenous vein. We believe that these select patients could be most suitable for totally endovascular transvenous bypass, which combines minimal invasiveness with effective revascularization of complex SFA lesions. This is especially important for long SFA occlusions, which have been somewhat underrepresented in earlier trials of endovascular SFA devices [14]. The results of our study must be evaluated in the context of long lesion length (mean 27.8 cm), which obviously has a significant impact on patency. There are recent publications analyzing the results of endovascular treatment of long TASC C and D SFA lesions. Labed et al. report their results with long (mean 29.5 cm) SFA lesions—mainly chronic total occlusions (85.9%)—treated with bare metal stents. They show similar 12-month but higher 24-month primary patency rates of 66.6% and 60.9%, respectively [15]. A recent multicenter trial has also reported their findings after stenting long (median stented length 252mm) femoropoliteal TASC C and D lesions. At 12 months, their primary patency rate is 67.0 ± 3.3%, which corresponds with our results and those reported by Laben et al. They did not report 24-month data [16].

The reported 1-year primary patency for open femoropopliteal bypass in a meta-analysis was 72% compared to endovascular revascularization, which had a lower primary patency of 62% at 1-year. This finding was also noted at the 2-year and 3-year follow-ups in the entire study population; however, if only randomized controlled trials were evaluated, 2-year and 3-year primary patency was similar between open and endovascular groups. Our results show a primary patency of 72.2% at 1 year which is similar to the open bypass reported in a meta-analysis [4]. One important aspect is lesion length and characteristics. The majority of studies in this meta-analysis did not define lesions’ length and did not specify how many chronic total occlusions were treated, whereas 2 of the studies reported lesion length <25 cm. In our study population, mean lesion length was 27.8 cm. This points to certain limitations in these trials and comparison between them, because lesion characteristics can strongly influence patency. Another group reported their data in consecutive patients with SFA lesions and Rutherford 3–6 disease who received either open or endovascular SFA revascularization. In their experience, 4-year primary patency was 69% for open and 66% for endovascular interventions [14], which is significantly better than our results for transvenous bypass (43.8% at 3 years). However, they only had 20% of TASC C-D lesions in the endovascular group and 65% in the open surgery group; thus, superior patency can partly be explained by less diseased SFA.

Direct and accurate comparisons of femoropopliteal bypass patency rates have certain inherent difficulties, such as disparities in conduit and surgical technique that is being used, inflow and outflow vessel conditions, antiplatelet regiments, and post-operative follow-up. While vein graft is universally considered to be superior to synthetic grafts, there are discrepancies in the proposed preference for Dacron vs. PTFE grafts, with latest reports suggesting the advantage of using Dacron [17]. The primary patency rates for an above-knee Dacron bypass are reported to be 80% at 12 months, 75% at 24 months, and 70% at 36 months [17]. However, the results of the data from several publications vary from no difference between graft materials to significantly lower 36-month primary patency of 46% for PTFE [17]. Our results suggest similar primary patency compared to PTFA grafts but lower rates compared to Dacron.

Based on our early experience, transvenous endovascular femoropopliteal bypass is also amenable to open and endovascular thrombectomy in case a complete graft thrombosis occurs. The majority of cases did not require repeated revascularization, probably because of sufficient collateral vessels. In cases where thrombectomy was clinically indicated, revascularization was uniformly successful and patency was maintained for a median of 16 months. This provided significantly higher secondary patency rates compared to primary patency rates. It must be noted that successful thrombectomy of bypass still jeopardizes run-off vessels due to distal embolization, which might produce detrimental effects in the long term.

When assessing our patency results, it becomes evident that stringent post-operative follow-up is vitally important to improve the durability of the transvenous bypass. Patency can be markedly increased by control Duplex ultrasound and subsequent balloon angioplasty of significant stenosis that could otherwise lead to graft failure. Close contact with the treating vascular surgeon and compliance with follow-up recommendations are paramount to sustain graft function and achieve a 66.3% 3-year primary-assisted patency rate, which is comparable with the aforementioned Dacron bypass primary patency results. As previously mentioned, due to several factors, it could be more accurate to compare patency rates between different bypass techniques within a particular center, instead of comparing results from different centers, healthcare systems, and perioperative care settings. Our research group is currently working on such a comparison in our center.

## 5. Conclusions

Transvenous endovascular femoropopliteal bypass is a viable option for selected cases who either lack sufficient saphenous vein, have long SFA lesions, or have comorbidities that increase the risk of open surgery.

This study had the following limitations included a small sample of study patients, and a relatively small number of study participants who had tissue loss, coronary artery disease, previous vascular interventions, diabetes, and chronic kidney disease. The study did not have a control group to compare patency with open surgical bypass. Our cohort consisted mainly of men who were relatively younger. Because of the relatively small number of subjects, we cannot strongly point towards factors that influenced patency.

## Figures and Tables

**Figure 1 medicina-59-00462-f001:**
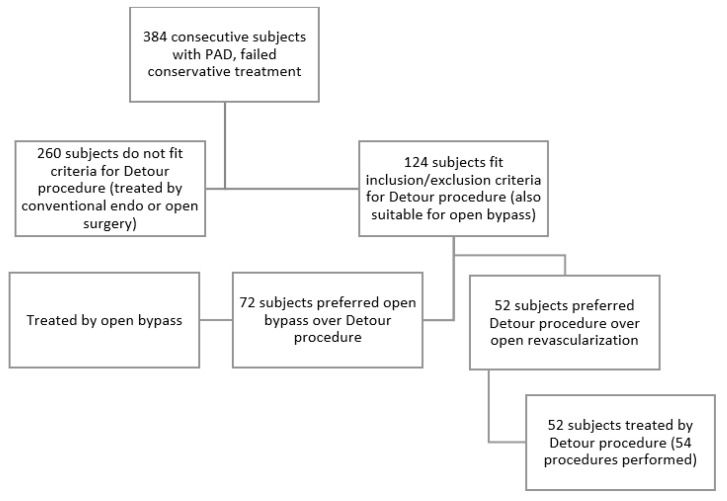
Flow chart depicting patient treatment and selection protocol.

**Figure 2 medicina-59-00462-f002:**
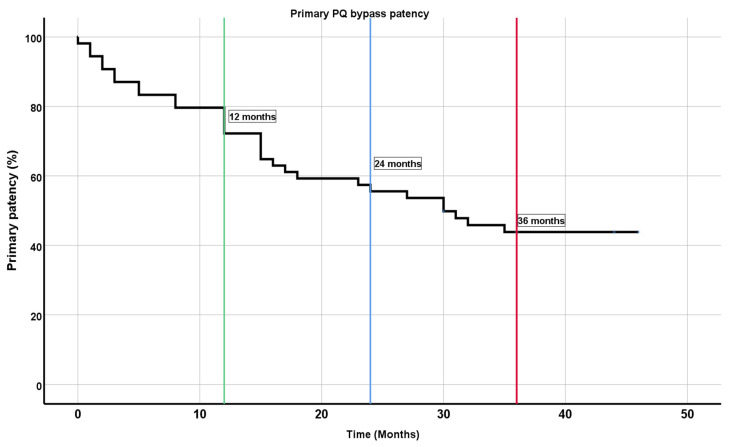
Kaplan–Meier plot for primary patency at 1 year (green line), 2 years (blue line), and 3 years (red line). PQ bypass, percutaneous endovascular transvenous femoropopliteal bypass.

**Figure 3 medicina-59-00462-f003:**
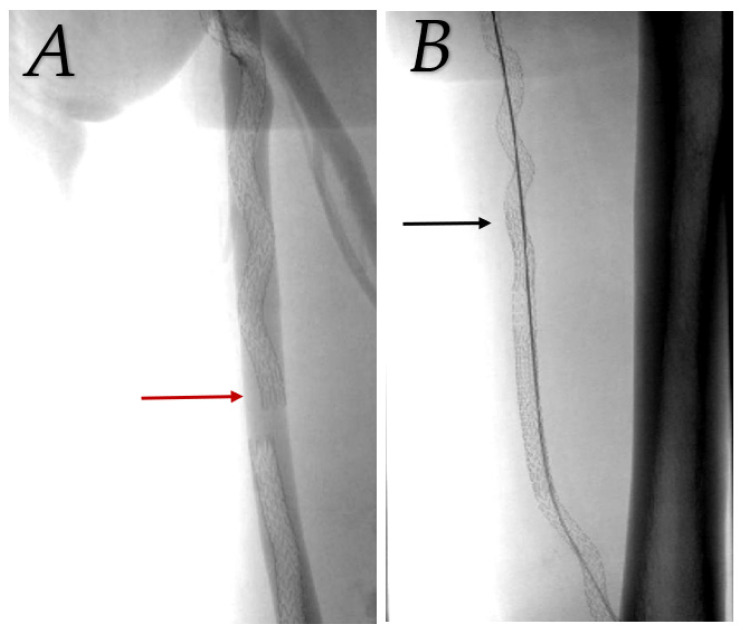
Detachment of stent grafts due to insufficient overlap (red arrow), arterial flow into femoral vein (**A**). Treated with bridging stent graft (black arrow, image (**B**)).

**Figure 4 medicina-59-00462-f004:**
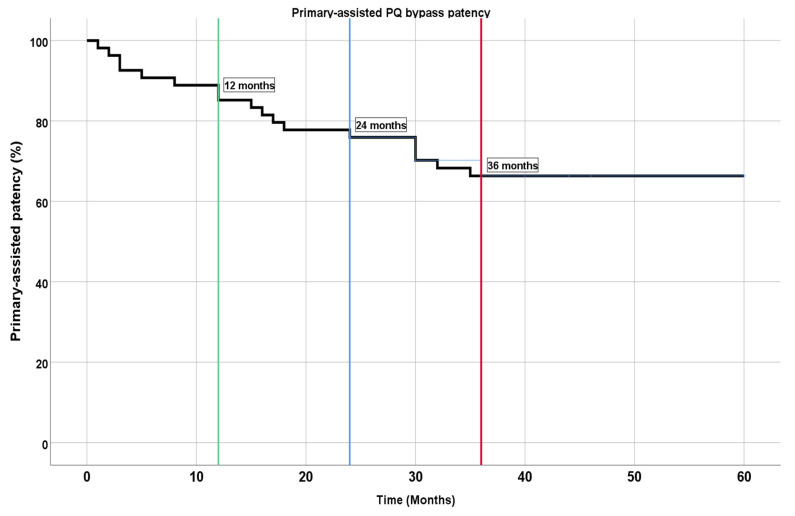
Kaplan–Meier plot for primary-assisted patency at 1 year (green line), 2 years (blue line), and 3 years (red line).

**Figure 5 medicina-59-00462-f005:**
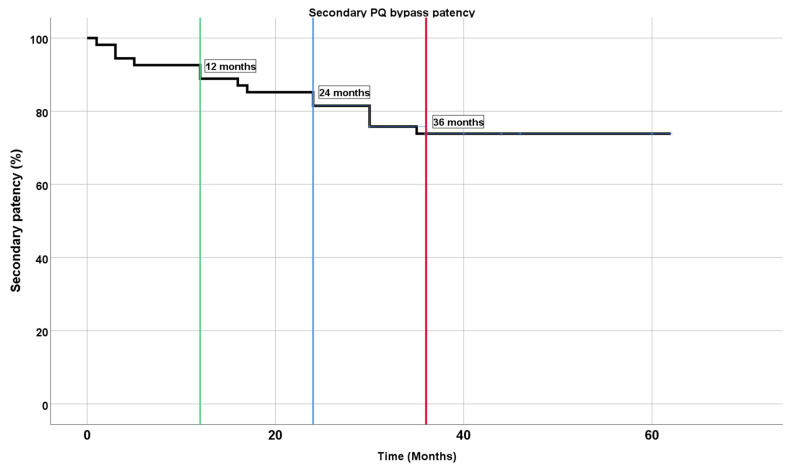
Kaplan–Meier plot for secondary patency at 1 year (green line), 2 years (blue line), and 3 years (red line).

**Table 1 medicina-59-00462-t001:** Study population and lesion characteristics.

**Age**	64.9 Years (SD = 8.3).
**Male**	92.3%
**Rutherford category**	3—84.6% 4—9.6% 5—5.8%
**Comorbidities**	Hypertension–88.4% Smoking—83.3% Hyperlipidemia—48.1% Coronary artery disease—28.8% Diabetes mellitus—25.9% Chronic kidney disease (I-II)—11.5% Previous peripheral interventions—25%
**ABI at baseline**	0.62 (SD = 0.16)
**SFA lesions’ length (cm, mean)**	27.8 cm (SD = 4.8)
**Chronic total occlusions**	94%
**Calcification**	Mild—17% Moderate—21% Severe—62%
**Run-off vessels**	1—5.8% 2—38.5% 3—55.7%
**Number of stent grafts implanted**	2—59.3% 3—38.8% 4—1.9%

**Table 2 medicina-59-00462-t002:** Patency data summary at different time points throughout follow-up period.

Time (Months)	Primary Patency (%)	Primary-Assisted Patency (%)	Secondary Patency (%)
12	72.2	88.9	92.6
24 36	55.6 43.8	77.8 66.3	85.2 73.9

## Data Availability

Not applicable.

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
