# Peer review of "Three-Year Patency Results following Endovascular Transvenous Femoropopliteal Bypass"

_medicina, 2023, doi:10.3390/medicina59030462_

Round 1

Reviewer 1 Report

Major aspects

The main concern for this paper consist of its single arm aspect, the absence of an traditional endovascular or open arm for comparison strongly reduces the interest, this because that in the literature there are many papers (trials, cohort study) mainly single arm, reporting data on the DETOURE bypass. In all these papers the conclusion is that it is safe, feasible and a viable alternative which are consistent with the conclusion in this paper.

The second criticism, consist of the fact that most of the patients in this study were claudicants. In my point of view, it seems a bit hazardous offering such a non-conventional procedure to claudicants (84,6%). Claudicants, according to the current literature, generally should be male, younger and with less complex lesions easily treatable by conventional angioplasty/stent. The cohort in this paper are in line with the literature as the authors report 92% males and a mean age of 64,9.  Have these patients undergone exercise rehabilitation? Have they attempt an endovascular approach as recommend by guide lines?  

Furthermore, authors conclude: Transvenous endovascular femoropopliteal bypass is a viable option for selected patients who lack adequate saphenous vein or have contraindications to  open femoropopliteal bypass. What kind of contraindications they refer to? Their cohort  is made of claudicants, relatively young and only 28,8% with CAD.

Minor aspects.

Abstract

The term novel technique. I don’t think it can be defined as a novel technique by now. In the literature, the first case series has been performed in 2014. The authors of this paper state the first procedure in 2015 (period of study 2015-2019)

Methods

-         I think that the information in the lines 61 -66 should stay in the Result-section

-         The information in the line 85-87, which seems to be the aim of this study, in my point of view, should stay in the introduction section.

Results.

-         No information is reported in this  paper in regard to the lesion apart from the  length. Were they CTO? What about the calcium component?

-         Which type of stent-graft were used, how many for a single procedure? Can you comment briefly about the cost of a single procedure, even though it is not the aim of the paper.

Discussion

The introductive part in this section (lines 137-167) is too much long, more than half of the whole the discussion. The authors report mainly data on the open bypass. Moreover they made even a statement “We believe that these select patients could be most suitable for totally endovascular transvenous bypass, which combines minimal invasiveness with effective revascularization of complex SFA lesions” referring to patients with tissue loss when their cohort includes  only  5 patients with Rutherford 5 and patients with Rutherford 6 were excluded from the study.

Conclusion

Too long

Nothing new to the current literature

Limitation. The authors should state the limitations of their study, such as the small sample of patients. 

Author Response

On behalf of the author collective, I thank you for your remarks!

  1. “The main concern for this paper consist of its single arm aspect, the absence of an traditional endovascular or open arm for comparison strongly reduces the interest, this because that in the literature there are many papers (trials, cohort study) mainly single arm, reporting data on the DETOURE bypass. In all these papers the conclusion is that it is safe, feasible and a viable alternative which are consistent with the conclusion in this paper.”

I agree with you that the single arm design does not allow a direct comparison between different methods. The scope of this particular research was to assess the patency of this revascularization technique. Further study is necessary to directly compare this with open approach, and we have started gathering data to perform this.

  1. “The second criticism, consist of the fact that most of the patients in this study were claudicants. In my point of view, it seems a bit hazardous offering such a non-conventional procedure to claudicants (84,6%). Claudicants, according to the current literature, generally should be male, younger and with less complex lesions easily treatable by conventional angioplasty/stent. The cohort in this paper are in line with the literature as the authors report 92% males and a mean age of 64,9.  Have these patients undergone exercise rehabilitation? Have they attempt an endovascular approach as recommend by guide lines?”

This is absolutely true that most patients with claudication are amenable to conventional angioplasty (if conservative treatment fails), which is also how the majority of these patients are treated in our center. Part of our standard of care is to undergo three months of exercise treatment before considering surgical treatment for claudicants, therefore, only patients who failed conservative treatment were offered a surgical treatment. In addition, both open and endovascular treatment was discussed with patients. Subjects were enrolled in this study according to a study protocol that was approved by regulatory and ethics commissions in several European countries, and as per protocol, we included patients who were eligible for both open and endovascular treatment. As these procedures were performed in a study setting with the effort to evaluate a relatively new device and lesions were long and mostly chronic total occlusions, conventional angioplasty with stent was not performed before this procedure.

  1. “Furthermore, authors conclude: Transvenous endovascular femoropopliteal bypass is a viable option for selected patients who lack adequate saphenous vein or have contraindications to  open femoropopliteal bypass. What kind of contraindications they refer to? Their cohort  is made of claudicants, relatively young and only 28,8% with CAD”

We have removed the words “contraindications to open surgery” from the manuscript. One of the exclusion criteria in this study was severe coronary artery disease with recent interventions or planned coronary angioplasty following peripheral intervention, therefore, the prevalence of coronary artery disease was 28.8% in our cohort.

  1. “The term novel technique. I don’t think it can be defined as a novel technique by now. In the literature, the first case series has been performed in 2014. The authors of this paper state the first procedure in 2015 (period of study 2015-2019)”

As you pointed out, “novel” might not be the best word, therefore, “novel” has been replaced by “relatively new”. 12-month results of Detour II trial were reported in the June of 2022, thus in authors opinion there is not an abundance of publications and information concerning various aspects of this device.

  1. “The information in the line 85-87, which seems to be the aim of this study, in my point of view, should stay in the introduction section.”

Lines 85-87 moved to introduction section as recommended by reviewer.

  1. “No information is reported in this  paper in regard to the lesion apart from the  length. Were they CTO? What about the calcium component?”

Information about prevalence of CTO and calcification of lesions added to result section in Table 1.

  1. “Which type of stent-graft were used, how many for a single procedure? Can you comment briefly about the cost of a single procedure, even though it is not the aim of the paper.”

As per your request and request by other reviewers, we have added a chapter regarding the procedure, type and number of stent grafts used in “Materials and methods” section.

  1. “Can you comment briefly about the cost of a single procedure, even though it is not the aim of the paper”

Since this device is not commercially available and procedures were done for study purposes, cost of a single procedure is currently unknown

  1. “Moreover they made even a statement “We believe that these select patients could be most suitable for totally endovascular transvenous bypass, which combines minimal invasiveness with effective revascularization of complex SFA lesions”referring to patients with tissue loss when their cohort includes  only  5 patients with Rutherford 5 and patients with Rutherford 6 were excluded from the study.”

The statement you refer to is meant about patients with tissue loss, who have a higher risk of conventional bypass graft infection, AND inadequate saphenous vein AND long, heavily calcified lesions (endovascular revascularization is preferable due to risk for infection, but conventional endovascular options are less effective for long and severely calcified lesions).

  1. “The introductive part in this section (lines 137-167) is too much long, more than half of the whole the discussion. The authors report mainly data on the open bypass”

Discussion is corrected but certain sentences kept to balance requests from other reviewers, who suggest a richer discussion

  1. “Too long. Nothing new to the current literature. Limitation. The authors should state the limitations of their study, such as the small sample of patients.”

Conclusion is shortened as you recommended, and we have added “limitations of this study” in conclusions

Reviewer 2 Report

Dear authors,

many thanks for submitting your manuscript entitled "Three-year patency results following endovascular transvenous femoropopliteal bypass" in Medicine Journal. The paper presents the single-centre experience with mid-term results of the Detour technique in treating infrainguinal femoropopliteal disease. Although the results are quite appealing for this technique, the manuscript contains a couple of issues that require the authors' attention. The major ones are labeled with asterix.

Introduction: 

No major remarks

* Methodology

* Line 72-74 does this mean these were patients not amenable to open bypass or standard transarterial recanalization? I still do not understand how did you select patients for this procedure. This has to be clear. Please depict the selection criteria for this procedure and treatment protocol in your institution with one flowchart.

* Line 80,81 - 36 months for dual antiplatelet therapy? Is not this too long? According to the latest ESVS guidelines DAPT can be considered for 1-6 months. How many bleeding complications (minor and major) did you have during the follow-up?

* Results:

* It would be nice to at least perform a univariate analysis and identify the factors influencing your outcomes.

* Can you list other comorbidities, such as CKD, previous CVI, HLP, etc?

* Can you please show a bit more details on the lesion characteristics: occlusion length/percentage, calcification, crural artery status, etc.

* I miss the procedure, postoperative details, complications, venous patency, etc.

- Line 100/101, 108-111, 123-125: you can insert these definitions in the methodology

* Please calculate the follow-up index for your study participants according to the instruction from the following paper: von Allmen RS, Weiss S, Tevaearai HT, Kuemmerli C, Tinner C, Carrel TP, Schmidli J, Dick F. Completeness of Follow-Up Determines Validity of Study Findings: Results of a Prospective Repeated Measures Cohort Study. PLoS One. 2015 Oct 15;10(10):e0140817. I am fearing that you have a significant attrition bias in your study.

Discussion: 

Please make it richer and more appealing for the reader by comparing this technique with the results of standard transarterial endovascular and open bypass techniques. 

Conclusion:

Mention only what comes from your results, not your opinion. Lines 189-197 are unnecessary. 

Author Response

On behalf of the author collective, I thank you for your remarks!

  1. “Line 72-74 does this mean these were patients not amenable to open bypass or standard transarterial recanalization? I still do not understand how did you select patients for this procedure. This has to be clear. Please depict the selection criteria for this procedure and treatment protocol in your institution with one flowchart”

We thank you for this suggestion. Because of the single arm design of this study, we did not add a patient selection flow-chart. Instead, we have clarified your question with additional statements in methodology section of this revised version. If the editor decides that it’s necessary to have such a flow-chart, we will definitely add it.

  1. “Line 80,81 - 36 months for dual antiplatelet therapy? Is not this too long? According to the latest ESVS guidelines DAPT can be considered for 1-6 months. How many bleeding complications (minor and major) did you have during the follow-up?”

Due to the fact that this was an investigational device and at that time it was not clear what the most optimal antiplatelet therapy regimen is for this particular device, we continued DAPT throughout the follow-up period. Furthermore, several graft thrombosis occurred after patients temporarily stopped antiplatelet therapy for different reasons, therefore, we decided we should continue DAPT. Two patients had minor bleeding. We did not see major bleeding episodes during follow-up.

  1. “It would be nice to at least perform a univariate analysis and identify the factors influencing your outcomes”

Thank you for this remark. One of the factors we noticed to be associated with graft patency was discontinuation of DAPT, but given the relatively small number of subjects, we can not strongly point towards other factors.

  1. “Can you list other comorbidities, such as CKD, previous CVI, HLP, etc –“

We have added more comorbidities to Table 1.

  1. “Can you please show a bit more details on the lesion characteristics: occlusion length/percentage, calcification, crural artery status, etc.”

Thank you for this request, we have added this information to Table 1

  1. “I miss the procedure, postoperative details, complications, venous patency”

As requested by reviewers, we have added a chapter regarding the procedure, type and number of stent grafts used. The rest of this information has been published previously, please see references [5-6].

  1. “Line 100/101, 108-111, 123-125: you can insert these definitions in the methodology”

We have done this.

  1. “Please calculate the follow-up index for your study participants according to the instruction from the following paper: von Allmen RS, Weiss S, Tevaearai HT, Kuemmerli C, Tinner C, Carrel TP, Schmidli J, Dick F. Completeness of Follow-Up Determines Validity of Study Findings: Results of a Prospective Repeated Measures Cohort Study. PLoS One. 2015 Oct 15;10(10):e0140817. I am fearing that you have a significant attrition bias in your study.””

Thank you for this suggestion and resource. Please correct me if I am wrong, but the article writes about (and later graphically demonstrates) a study with a fixed start and end date, where these discrepancies can and do appear. This differs from our study because we followed all and every patient for 36-months, irrespective of the start time for every particular patient. The article writes “Ideally, study findings should be based on complete follow-up information [12]. But in reality, it may be impracticable to follow every single study participant exactly to the study end date” which is exactly what we did in our study. “Therefore, studies should declare at least how complete their follow-up was”, which we have done in methodology section stating that “One patient was lost to follow-up at the start of the study, one patient died 8 months after the procedure of an unrelated cause, and one patient did not attend follow-up after 24-months.” Obviously, it is not a perfect follow up for all patients, but for all the remaining ones we did continue follow up for 36-months irrespective of events (for example, graft thrombosis). For now we leave this question in the discretion of the editor, but if you believe strongly that it is necessary, we will consult statisticians in this topic and calculate this index.

  1. “Please make it richer and more appealing for the reader by comparing this technique with the results of standard transarterial endovascular and open bypass techniques”

Chapter added in discussion with comparison of results between our cohort and traditional open and endovascular revascularization

Reviewer 3 Report

Thank you very much for the opportunity to review this valuable paper.

I learned a lot from your review of the Detour procedure, as it is a hot topic.

As you mentioned, there are pros and cons between open surgery and endovascular treatment for PAD.

I understood that under certain conditions this procedure is effective for the patient.

(1) Although this study only focused on the presence or absence of patency, it would have been nice to have clinical improvement as an outcome, such as improvement in claudication symptoms, improvement in ischemic symptoms, and so on.

I think it needs to be compared to surgical treatment. Therefore, it seemed that presenting post-treatment data would assert the effectiveness of this procedure.

(2) Removal of the intra-graft thrombus does not mean that peripheral circulation in that leg is restored. The peripheral circulation will certainly deteriorate due to peripheral embolization caused by the residual microthrombus that could not be retrieved.

Is there any result of evaluation of peripheral circulation in this case?

(3) How many cases of bilateral legs of the same person were included?

(4) There is little reference to the patient's background. I think the underlying disease should be clarified.

(5) Did this device cause any venous problems?

It takes courage, but I think that clarifying the negative aspects will contribute to research at other institutions in the future.

Author Response

On behalf of the author collective, I thank you for your remarks!

  • Although this study only focused on the presence or absence of patency, it would have been nice to have clinical improvement as an outcome, such as improvement in claudication symptoms, improvement in ischemic symptoms, and so on.

Thank you for your suggestions. It was not the primary focus of this particular study, therefore we did not include these data. As you requested, we have added a chapter in result section “Clinical arterial results”

  • Removal of the intra-graft thrombus does not mean that peripheral circulation in that leg is restored. The peripheral circulation will certainly deteriorate due to peripheral embolization caused by the residual microthrombus that could not be retrieved. Is there any result of evaluation of peripheral circulation in this case?

We absolutely agree that every thrombotic episode can have a detrimental effect on peripheral circulation and cause distal embolization. During the procedure we did a control angiography at the end of procedure, as well as ankle-brachial index measurements at every visit. We did see clinically non-significant distal embolization in few of these procedures. However, after open thrombectomy from graft we did not routinely perform angiography control, if it was not clinically indicated. 

  1. How many cases of bilateral legs of the same person were included?

We had two patients with bilateral disease who received Detour procedures

  1. Did this device cause any venous problems?

These data have been published separately, please see references [5-6]

Round 2

Reviewer 1 Report

The paper has been extensively improved and it can be accepted in the revised version. 

Author Response

Dear reviewer,

Thank you for your time and recommendations!

Reviewer 2 Report

Dear authors,

many thanks for resubmitting your article. The quality of the paper improved a lot with the insertion of additional data and enrichment of the discussion.

I would still like you to add an additional flowchart of all treated patients and FUI for included patients during the follow-up period of 36 months, but I leave this to the discretion of the editor.

Author Response

On behalf of the author collective, I thank you for your additional remarks!

As you suggested, we have added a flow-chart depicting patient selection process for Detour procedure after Methodology section with a reference in the text (lines 84-85). And as you also noted, let us leave this to the discretion of the editor.